# GIANT: Globally Improved Approximate Newton Method for Distributed Optimization

**Shusen Wang**
Stevens Institute of Technology
shusen.wang@stevens.edu

**Farbod Roosta-Khorasani**
University of Queensland
fred.roosta@uq.edu.au

**Peng Xu**
Stanford University
pengxu@stanford.edu

**Michael W. Mahoney**
University of California at Berkeley
mmahoney@stat.berkeley.edu

## Abstract

For distributed computing environment, we consider the empirical risk minimization problem and propose a distributed and communication-efficient Newton-type optimization method. At every iteration, each worker locally finds an Approximate NewTon (ANT) direction, which is sent to the main driver. The main driver, then, averages all the ANT directions received from workers to form a *Globally Improved ANT* (GIANT) direction. GIANT is highly communication efficient and naturally exploits the trade-offs between local computations and global communications in that more local computations result in fewer overall rounds of communications. Theoretically, we show that GIANT enjoys an improved convergence rate as compared with first-order methods and existing distributed Newton-type methods. Further, and in sharp contrast with many existing distributed Newton-type methods, as well as popular first-order methods, a highly advantageous practical feature of GIANT is that it only involves one tuning parameter. We conduct large-scale experiments on a computer cluster and, empirically, demonstrate the superior performance of GIANT.

## 1 Introduction

The large-scale nature of many modern "big-data" problems, arising routinely in science, engineering, financial markets, Internet and social media, etc., poses significant computational as well as storage challenges for machine learning procedures. For example, the scale of data gathered in many applications nowadays typically exceeds the memory capacity of a single machine, which, in turn, makes learning from data ever more challenging. In this light, several modern parallel (or distributed) computing architectures, e.g., MapReduce [4], Apache Spark [44, 19], GraphLab [14], and Parameter Server [11], have been designed to operate on and learn from data at massive scales. Despite the fact that, when compared to a single machine, distributed systems tremendously reduce the storage and (local) computational costs, the inevitable cost of communications across the network can often be the bottleneck of distributed computations. As a result, designing methods which can strike an appropriate balance between the cost of computations and that of communications are increasingly desired.

The desire to reduce communication costs is even more pronounced in the *federated learning* framework [8, 9, 1, 18, 37]. Similarly to typical settings of distributed computing, federated learning assumes data are distributed over a network across nodes that enjoy reasonable computational resources, e.g., mobile phones, wearable devices, and smart homes. However, the network has

severely limited bandwidth and high latency. As a result, it is imperative to reduce the communications between the center and a node or between two nodes. In such settings, the preferred methods are those which can perform expensive local computations with the aim of reducing the overall communications across the network.

Optimization algorithms designed for distributed setting are abundantly found in the literature. First-order methods, i.e, those that rely solely on gradient information, are often embarrassingly parallel and easy to implement. Examples of such methods include distributed variants of stochastic gradient descent (SGD) [17, 27, 47], accelerated SGD [35], variance reduction SGD [10, 28], stochastic coordinate descent methods [5, 13, 20, 31] and dual coordinate ascent algorithms [30, 43, 46]. The common denominator in all of these methods is that they significantly reduce the amount of local computation. But this blessing comes with an inevitable curse that they, in turn, may require a far greater number of iterations and hence, incur more communications overall. Indeed, as a result of their highly iterative nature, many of these first-order methods require several rounds of communications and, potentially, synchronizations in every iteration, and they must do so for many iterations. In a computer cluster, due to limitations on the network's bandwidth and latency and software system overhead, communications across the nodes can oftentimes be the critical bottleneck for the distributed optimization. Such overheads are increasingly exacerbated by the growing number of compute nodes in the network, limiting the scalability of any distributed optimization method that requires many communication-intensive iterations.

To remedy such drawbacks of high number of iterations for distributed optimization, communication-efficient second-order methods, i.e., those that, in addition to the gradient, incorporate curvature information, have also been recently considered [16, 36, 29, 45, 7, 15, 38]; see also Section 1.1. The common feature in all of these methods is that they intend to increase the local computations with the aim of reducing the overall iterations, and hence, lowering the communications. In other words, these methods are designed to perform as much local computation as possible before making any

Table 1: Commonly used notation.

| Notation | Definition |
|---|---|
| $n$ | total number of samples |
| $d$ | number of features (attributes) |
| $m$ | number of partitions |
| $f$ | objective function |
| $\gamma$ | regularization parameter |
| $\mathbf{w}_t$ | the variable at iteration $t$ |
| $\mathbf{w}^\star$ | the variable that minimizes $f$ |
| $\kappa$ | some condition number |

communications across the network. Pursuing similar objectives, in this paper, we propose a Globally Improved Approximate NewTon (GIANT) method and establish its improved theoretical convergence properties as compared with other similar second-order methods. We also showcase the superior empirical performance of GIANT through several numerical experiments.

The rest of this paper is organized as follows. Section 1.1 briefly reviews prior works most closely related to this paper. Section 1.2 gives a summary of our main contributions. The formal description of the distributed empirical risk minimization problem is given in Section 2, followed by the derivation of various steps of GIANT in Section 3. Section 4 presents the theoretical guarantees. The most commonly used notation is listed in Table 1. Due to the page limit, the readers can refer to the long version [41]; Section 5 provides a summary of our experiments. The proofs can be found in the long version [41].

Table 2: The number of communications (proportional to the number of iterations) required for the ridge regression problem. Here $\kappa$ is the condition number of the Hessian matrix, $\mu$ is the matrix coherence, and $\tilde{\mathcal{O}}$ conceals constants (analogous to $\mu$) and logarithmic factors.

| Method | #Iterations | Metric |
|---|---|---|
| GIANT [this work] | $t = \mathcal{O}\left( \frac{\log(d\kappa/\mathcal{E})}{\log(n/\mu dm)} \right)$ | $\|\mathbf{w}_t - \mathbf{w}^\star\|_2 \leq \mathcal{E}$ |
| DiSCO [45] | $t = \tilde{\mathcal{O}}\left( \frac{d\kappa^{1/2}m^{3/4}}{n^{3/4}} + \frac{\kappa^{1/2}m^{1/4}}{n^{1/4}} \log \frac{1}{\mathcal{E}} \right)$ | $f(\mathbf{w}_t) - f(\mathbf{w}^\star) \leq \mathcal{E}$ |
| DANE [36] | $t = \tilde{\mathcal{O}}\left( \frac{\kappa^2 m}{n} \log \frac{1}{\mathcal{E}} \right)$ | $f(\mathbf{w}_t) - f(\mathbf{w}^\star) \leq \mathcal{E}$ |
| AIDE [29] | $t = \tilde{\mathcal{O}}\left( \frac{\kappa^{1/2}m^{1/4}}{n^{1/4}} \log \frac{1}{\mathcal{E}} \right)$ | $f(\mathbf{w}_t) - f(\mathbf{w}^\star) \leq \mathcal{E}$ |
| CoCoA [38] | $t = \mathcal{O}\left( \left(n + \frac{1}{\gamma}\right) \log \frac{n}{\mathcal{E}} \right)$ | $f(\mathbf{w}_t) - f(\mathbf{w}^\star) \leq \mathcal{E}$ |
| AGD | $t = \mathcal{O}\left( \kappa^{1/2} \log \frac{d}{\mathcal{E}} \right)$ | $\|\mathbf{w}_t - \mathbf{w}^\star\|_2 \leq \mathcal{E}$ |

## 1.1 Related Work

Among the existing distributed second-order optimization methods, the most notably are DANE [36], AIDE [29], and DiSCO [45]. Another similar method is CoCoA [7, 15, 38], which is analogous to second-order methods in that it involves sub-problems which are local quadratic approximations to the dual objective function. However, despite the fact that CoCoA makes use of the smoothness condition, it does not exploit any explicit second-order information.

We can evaluate the theoretical properties the above-mentioned methods in light of comparison with optimal first-order methods, i.e., accelerated gradient descent (AGD) methods [22, 23]. It is because AGD methods are mostly embarrassingly parallel and can be regarded as the baseline for distributed optimization. Recall that AGD methods, being optimal in worst-case analysis sense [21], are guaranteed to convergence to $\mathcal{E}$-precision in $\mathcal{O}(\sqrt{\kappa} \log \frac{1}{\mathcal{E}})$ iterations [23], where $\kappa$ can be thought of as the condition number of the problem. Each iteration of AGD has two rounds of communications—broadcast or aggregation of a vector.

In Table 2, we compare the communication costs with other methods for the ridge regression problem: $\min_{\mathbf{w}} \frac{1}{n} \|\mathbf{X}\mathbf{w} - \mathbf{y}\|_2^2 + \gamma \|\mathbf{w}\|_2^2$.[1] The communication cost of GIANT has a mere logarithmic dependence on the condition number $\kappa$; in contrast, the other methods have at least a square root dependence on $\kappa$. Even if $\kappa$ is assumed to be small, say $\kappa = \mathcal{O}(\sqrt{n})$, which was made by [45], GIANT's bound is better than the compared methods regarding the dependence on the number of partitions, $m$.

Our GIANT method is motivated by the subsampled Newton method [33, 42, 25]. Later on, we realized that a similar idea has been proposed by DANE [36]; GIANT and DANE are identical for quadratic programming; they are different for the general convex problems. Nevertheless, we show better convergence bounds than DANE, even for quadratic programming. Our improvement over DANE is obtained by better bounds the Hessian matrix approximation and better analysis of convex optimization.

GIANT also bears a resemblance to FADL [16], but we show better convergence bounds. Mahajan *et al.* [16] has conducted comprehensive empirical comparisons among many distributed computing methods and concluded that the local quadratic approximation, which is very similar to GIANT, is the final method which they recommended.

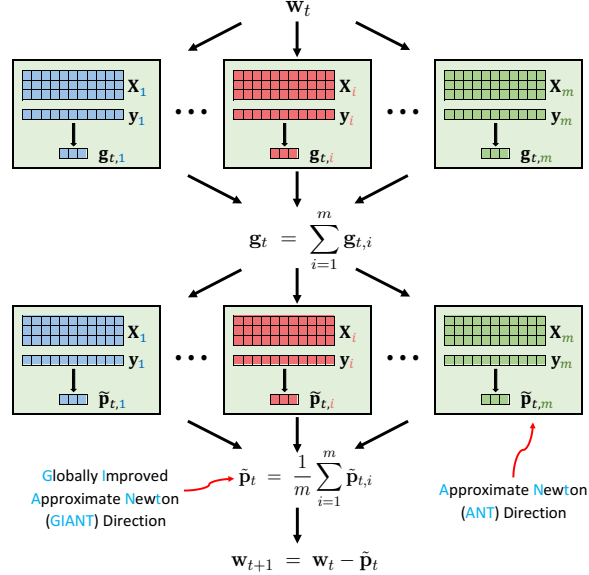

Figure 1: One iteration of GIANT. Here $\mathbf{X}$ and $\mathbf{y}$ are respectively the features and lables; $\mathbf{X}_i$ and $\mathbf{y}_i$ denotes the blocks of $\mathbf{X}$ and $\mathbf{y}$, respectively. Each one-to-all operation is a Broadcast and each all-to-one operation is a Reduce.

## 1.2 Contributions

In this paper, we consider the problem of empirical risk minimization involving smooth and strongly convex objective function (which is the same setting considered in prior works of DANE, AIDE, and DiSCO). In this context, we propose a Globally Improved Approximate NewTon (GIANT) method and establish its theoretical and empirical properties as follows.

● For quadratic objectives, we establish global convergence of GIANT. To attain a fixed precision, the number of iterations of GIANT (which is proportional to the communication complexity) has

a mere *logarithmic* dependence on the condition number. In contrast, the prior works have at least square root dependence. In fact, for quadratic problems, GIANT and DANE [36] can be shown to be identical. In this light, for such problems, our work improves upon the convergence of DANE.

• For more general problems, GIANT has *linear-quadratic convergence* in the vicinity of the optimal solution, which we refer to as "local convergence".[2] The advantage of GIANT mainly manifests in *big-data* regimes where there are many data points available. In other words, when the number of data points is much larger than the number of features, the theoretical convergence of GIANT enjoys significant improvement over other similar methods.

• In addition to theoretical features, GIANT also exhibits desirable practical advantages. For example, in sharp contrast with many existing distributed Newton-type methods, as well as popular first-order methods, GIANT only involves *one tuning parameter*, i.e., the maximal iterations of its sub-problem solvers, which makes GIANT easy to implement in practice. Furthermore, our experiments on a computer cluster show that GIANT consistently outperforms AGD, L-BFGS, and DANE.

## 2  Problem Formulation

In this paper, we consider the distributed variant of empirical risk minimization, a supervised-learning problem arising very often in machine learning and data analysis [34]. More specifically, let $\mathbf{x}_1, \cdots, \mathbf{x}_n \in \mathbb{R}^d$ be the input feature vectors and $y_1, \cdots, y_n \in \mathbb{R}$ be the corresponding response. The goal of supervised learning is to compute a model from the training data, which can be achieved by minimizing an empirical risk function, i.e.,

$$\min_{\mathbf{w} \in \mathbb{R}^d} \left\{ f(\mathbf{w}) \triangleq \frac{1}{n} \sum_{j=1}^n \ell_j(\mathbf{w}^T \mathbf{x}_j) + \frac{\gamma}{2} \|\mathbf{w}\|_2^2 \right\}, \tag{1}$$

where $\ell_j : \mathbb{R} \mapsto \mathbb{R}$ is convex, twice differentiable, and smooth. We further assume that $f$ is strongly convex, which in turn, implies the uniqueness of the minimizer of (1), denoted throughout the text by $\mathbf{w}^\star$. Note that $y_j$ is implicitly captured by $\ell_j$. Examples of the loss function, $\ell_j$, appearing in (1) include

$$\begin{aligned} \text{linear regression:} \quad & \ell_j(z_j) = \tfrac{1}{2}(z_j - y_j)^2, \\ \text{logistic regression:} \quad & \ell_j(z_j) = \log(1 + e^{-z_j y_j}). \end{aligned}$$

Suppose the $n$ feature vectors and loss functions $(\mathbf{x}_1, \ell_1), \cdots, (\mathbf{x}_n, \ell_n)$ are partitioned among $m$ worker machines. Let $s \triangleq n/m$ be the local sample size. Our theories require $s > d$; nevertheless, GIANT empirically works well for $s < d$.

We consider solving (1) in the regimes where $n \gg d$. We assume that the data points, $\{\mathbf{x}_i\}_{i=1}^n$ are partitioned among $m$ machines, with possible overlaps, such that the number of local data is larger than $d$. Otherwise, if $n \ll d$, we can consider the dual problem and partition features. If the dual problem is also decomposable, smooth, strongly convex, and unconstrained, e.g., ridge regression, then our approach directly applies.

## 3  Algorithm Description

In this section, we present the algorithm derivation and complexity analysis. GIANT is a centralized and synchronous method; one iteration of GIANT is depicted in Figure 1. The key idea of GIANT is avoiding forming of the exact Hessian matrices $\mathbf{H}_t \in \mathbb{R}^{d \times d}$ in order to avoid expensive communications.

## 3.1 Gradient and Hessian

GIANT iterations require the exact gradient, which in the $t$-th iteration, can be written as

$$\mathbf{g}_t = \nabla f(\mathbf{w}_t) = \frac{1}{n} \sum_{j=1}^{n} \ell_j'(\mathbf{w}_t^T \mathbf{x}_j)\, \mathbf{x}_j + \gamma \mathbf{w}_t \in \mathbb{R}^d. \qquad (2)$$

The gradient, $\mathbf{g}_t$ can be computed, embarrassingly, in parallel. The driver Broadcasts $\mathbf{w}_t$ to all the worker machines. Each machine then uses its own $\{(\mathbf{x}_j, \ell_j)\}$ to compute its local gradient. Subsequently, the driver performs a Reduce operation to sum up the local gradients and get $\mathbf{g}_t$. The per-iteration communication complexity is $\tilde{\mathcal{O}}(d)$ words, where $\tilde{\mathcal{O}}$ hides the dependence on $m$ (which can be $m$ or $\log m$, depending on the network structure).

More specifically, in the $t$-th iteration, the Hessian matrix at $\mathbf{w}_t \in \mathbb{R}^d$ can be written as

$$\mathbf{H}_t = \nabla^2 f(\mathbf{w}_t) = \frac{1}{n} \sum_{j=1}^{n} \ell_j''(\mathbf{w}_t^T \mathbf{x}_j) \cdot \mathbf{x}_j \mathbf{x}_j^T + \gamma \mathbf{I}_d. \qquad (3)$$

To compute the exact Hessian, the driver must aggregate the $m$ local Hessian matrices (each of size $d \times d$) by one Reduce operation, which has $\tilde{\mathcal{O}}(d^2)$ communication complexity and is obviously impractical when $d$ is thousands. The Hessian approximation developed in this paper has a mere $\tilde{\mathcal{O}}(d)$ communication complexity which is the same to the first-order methods.

## 3.2 Approximate NewTon (ANT) Directions

Assume each worker machine locally holds $s$ random samples drawn from $\{(\mathbf{x}_j, \ell_j)\}_{j=1}^{n}$.[3] Let $\mathcal{J}_i$ be the set containing the indices of the samples held by the $i$-th machine, and $s = |\mathcal{J}_i|$ denote its size. Each worker machine can use its local samples to form a local Hessian matrix

$$\widetilde{\mathbf{H}}_{t,i} = \frac{1}{s} \sum_{j \in \mathcal{J}_i} \ell_j''(\mathbf{w}_t^T \mathbf{x}_j) \cdot \mathbf{x}_j \mathbf{x}_j^T + \gamma \mathbf{I}_d.$$

Clearly, $\mathbb{E}[\widetilde{\mathbf{H}}_{t,i}] = \mathbf{H}_t$. We define the Approximate NewTon (ANT) direction by $\tilde{\mathbf{p}}_{t,i} = \widetilde{\mathbf{H}}_{t,i}^{-1} \mathbf{g}_t$. The cost of computing the ANT direction $\tilde{\mathbf{p}}_{t,i}$ in this way, involves $\mathcal{O}(sd^2)$ time to form the $d \times d$ dense matrix $\widetilde{\mathbf{H}}_{t,i}$ and $\mathcal{O}(d^3)$ to invert it.

To reduce the computational cost, we opt to compute the ANT direction by the conjugate gradient (CG) method [24]. Let $\mathbf{a}_j = \sqrt{\ell_j''(\mathbf{w}_t^T \mathbf{x}_j)} \cdot \mathbf{x}_j \in \mathbb{R}^d$,

$$\mathbf{A}_t = [\mathbf{a}_1^T; \cdots ; \mathbf{a}_n^T] \in \mathbb{R}^{n \times d}, \qquad (4)$$

and $\mathbf{A}_{t,i} \in \mathbb{R}^{s \times d}$ contain the rows of $\mathbf{A}_t$ indexed by the set $\mathcal{J}_i$. Using the matrix notation, we can write the local Hessian matrix as

$$\widetilde{\mathbf{H}}_{t,i} = \tfrac{1}{s} \mathbf{A}_{t,i}^T \mathbf{A}_{t,i} + \gamma \mathbf{I}_d. \qquad (5)$$

Employing CG, it is unnecessary to explicitly form $\widetilde{\mathbf{H}}_{t,i}$. Indeed, one can simply approximately solve

$$\left( \tfrac{1}{s} \mathbf{A}_{t,i}^T \mathbf{A}_{t,i} + \gamma \mathbf{I}_d \right) \mathbf{p} = \mathbf{g}_t \qquad (6)$$

in a "Hessian-free" manner, i.e., by employing only Hessian-vector products in CG iterations. In each round of GIANT, the local computational cost of a worker machine is $\mathcal{O}(q \cdot \text{nnz}(\mathbf{A}_{t,i}))$, where $q$ is the number of CG iterations specified by the users and typically set to tens.

## 3.3 Globally Improved ANT (GIANT) Direction

Using random matrix concentration, we can show that for sufficiently large $s$, the local Hessian matrix $\widetilde{\mathbf{H}}_{t,i}$ is a spectral approximation to $\mathbf{H}_t$. Now let $\tilde{\mathbf{p}}_{t,i}$ be an ANT direction. The Globally Improved ANT (GIANT) direction is defined as

$$\tilde{\mathbf{p}}_t \;=\; \frac{1}{m}\sum_{i=1}^{m}\tilde{\mathbf{p}}_{t,i} \;=\; \frac{1}{m}\sum_{i=1}^{m}\widetilde{\mathbf{H}}_{t,i}^{-1}\mathbf{g}_t \;=\; \widetilde{\mathbf{H}}_t^{-1}\mathbf{g}_t. \tag{7}$$

Interestingly, here $\widetilde{\mathbf{H}}_t$ is the *harmonic mean* defined as $\widetilde{\mathbf{H}}_t \triangleq (\frac{1}{m}\sum_{i=1}^{m}\widetilde{\mathbf{H}}_{t,i}^{-1})^{-1}$, whereas the true Hessian $\mathbf{H}_t$ is the *arithmetic mean* defined as $\mathbf{H}_t \triangleq \frac{1}{m}\sum_{i=1}^{m}\widetilde{\mathbf{H}}_{t,i}$. If the data is incoherent, that is, the "information" is spread-out rather than concentrated to a small fraction of samples, then the harmonic mean and the arithmetic mean are very close to each other, and thereby the GIANT direction $\tilde{\mathbf{p}}_t = \widetilde{\mathbf{H}}^{-1}\mathbf{g}_t$ very well approximates the true Newton direction $\mathbf{H}^{-1}\mathbf{g}_t$. This is the intuition of our global improvement.

The motivation of using the harmonic mean, $\widetilde{\mathbf{H}}_t$, to approximate the arithmetic mean (the true Hessian matrix), $\mathbf{H}_t$, is the communication cost. Computing the arithmetic mean $\mathbf{H}_t \triangleq \frac{1}{m}\sum_{i=1}^{m}\widetilde{\mathbf{H}}_{t,i}$ would require the communication of $d \times d$ matrices which is very expensive. In contrast, computing $\tilde{\mathbf{p}}_t$ merely requires the communication of $d$-dimensional vectors.

## 3.4 Time and Communication Complexities

For each worker machine, the per-iteration time complexity is $\mathcal{O}(sdq)$, where $s$ is the local sample size and $q$ is the number of CG iterations for (approximately) solving (6). (See Proposition 5 for the setting of $q$.) If the feature matrix $\mathbf{X} \in \mathbb{R}^{n \times d}$ has a sparsity of $\varrho = \mathrm{nnz}(\mathbf{X})/(nd) < 1$, the expected per-iteration time complexity is then $\mathcal{O}(\varrho sdq)$.

Each iteration of GIANT has four rounds of communications: two Broadcast for sending and two Reduce for aggregating some $d$-dimensional vector. If the communication is in a tree fashion, the per-iteration communication complexity is then $\tilde{\mathcal{O}}(d)$ words, where $\tilde{\mathcal{O}}$ hides the factor involving $m$ which can be $m$ or $\log m$. In contrast, the naive Newton's method has $\tilde{\mathcal{O}}(d^2)$ communication complexity, because the system sends and receives $d \times d$ Hessian matrices.

# 4 Theoretical Analysis

In this section, we formally present the convergence guarantees of GIANT. Section 4.1 focuses on quadratic loss and treats the global convergence of GIANT. This is then followed by local convergence properties of GIANT for more general non-quadratic loss in Section 4.2. For the results of Sections 4.1 and 4.2, we require that the local linear system to obtain the local Newton direction is solved exactly. Section 4.3 then relaxes this requirement to allow for inexactness in the solution, and establishes similar convergence rates as those of exact variants.

For our analysis here, we frequently make use of the notion of *matrix row coherence*, defined as follows. Such a notation has been used in compressed sensing [3], matrix completion [2], and randomized linear algebra [6, 40, 39].

**Definition 1** (Coherence). *Let $\mathbf{A} \in \mathbb{R}^{n \times d}$ be any matrix and $\mathbf{U} \in \mathbb{R}^{n \times d}$ be its column orthonormal bases. The row coherence of $\mathbf{A}$ is $\mu(\mathbf{A}) = \frac{n}{d}\max_j \|\mathbf{u}_j\|_2^2 \in [1, \frac{n}{d}]$.*

**Remark 1.** *Our work assumes $\mathbf{A}_t \in \mathbb{R}^{n \times d}$, which is defined in (4), is incoherent, namely $\mu(\mathbf{A}_t)$ is small. The prior works, DANE, AIDE, and DiSCO, did not use the notation of* incoherence*; instead, they assume $\nabla_{\mathbf{w}}^2 l_j(\mathbf{w}^T \mathbf{x}_j)|_{\mathbf{w}=\mathbf{w}_t} = \mathbf{a}_j \mathbf{a}_j^T$ is upper bounded for all $j \in [n]$ and $\mathbf{w}_t \in \mathbb{R}^d$, where $\mathbf{a}_j \in \mathbb{R}^d$ is the $j$-th row of $\mathbf{A}_t$. Such an assumption is different from but has similar implication as our incoherence assumption; under either of the two assumptions, it can be shown that the Hessian matrix can be approximated using a subset of samples selected uniformly at random.*

## 4.1 Quadratic Loss

In this section, we consider a special case of (1) with $\ell_i(z) = (z - y_i)^2/2$, i.e., the quadratic optimization problems:

$$f(\mathbf{w}) = \frac{1}{2n}\|\mathbf{X}\mathbf{w} - \mathbf{y}\|_2^2 + \frac{\gamma}{2}\|\mathbf{w}\|_2^2. \tag{8}$$

The Hessian matrix is given as $\nabla^2 f(\mathbf{w}) = \frac{1}{n}\mathbf{X}^T\mathbf{X} + \gamma\mathbf{I}_d$, which does not depend on $\mathbf{w}$. Theorem 1 describes the convergence of the error in the iterates, i.e., $\boldsymbol{\Delta}_t \triangleq \mathbf{w}_t - \mathbf{w}^\star$.

**Theorem 1.** *Let $\mu$ be the row coherence of $\mathbf{X} \in \mathbb{R}^{n \times d}$ and $m$ be the number of partitions. Assume the local sample size satisfies $s \geq \frac{3\mu d}{\eta^2}\log\frac{md}{\delta}$ for some $\eta, \delta \in (0, 1)$. It holds with probability $1 - \delta$ that*

$$\|\boldsymbol{\Delta}_t\|_2 \leq \alpha^t \sqrt{\kappa} \|\boldsymbol{\Delta}_0\|_2,$$

*where $\alpha = \frac{\eta}{\sqrt{m}} + \eta^2$ and $\kappa$ is the condition number of $\nabla^2 f(\mathbf{w}) = \frac{1}{n}\mathbf{X}^T\mathbf{X} + \gamma\mathbf{I}_d$.*

**Remark 2.** *The theorem can be interpreted in the this way. Assume the total number of samples, $n$, is at least $3\mu dm\log(md)$. Then*

$$\|\boldsymbol{\Delta}_t\|_2 \leq \left(\frac{3\mu dm\log(md/\delta)}{n} + \sqrt{\frac{3\mu d\log(md/\delta)}{n}}\right)^t \sqrt{\kappa}\|\boldsymbol{\Delta}_0\|_2$$

*holds with probability at least $1 - \delta$.*

*If the total number of samples, $n$, is substantially bigger than $\mu dm$, then GIANT converges in a very small number of iterations. Furthermore, to reach a fixed precision, say $\|\boldsymbol{\Delta}_t\|_2 \leq \mathcal{E}$, the number of iterations, $t$, has a mere logarithmic dependence on the condition number, $\kappa$.*

## 4.2 General Smooth Loss

For more general (not necessarily quadratic) but smooth loss, GIANT has linear-quadratic local convergence, which is formally stated in Theorem 2 and Corollary 3. Let $\mathbf{H}^\star = \nabla^2 f(\mathbf{w}^\star)$ and $\mathbf{H}_t = \nabla^2 f(\mathbf{w}_t)$. For this general case, we assume the Hessian is $L$-Lipschitz, which is a standard assumption in analyzing second-order methods.

**Assumption 1.** *The Hessian matrix is $L$-Lipschitz continuous, i.e., $\left\|\nabla^2 f(\mathbf{w}) - \nabla^2 f(\mathbf{w}')\right\|_2 \leq L\|\mathbf{w} - \mathbf{w}'\|_2$, for all $\mathbf{w}$ and $\mathbf{w}'$.*

Theorem 2 establishes the linear-quadratic convergence of $\boldsymbol{\Delta}_t \triangleq \mathbf{w}_t - \mathbf{w}^\star$. We remind that $\mathbf{A}_t \in \mathbb{R}^{n \times d}$ is defined in (4) (thus $\mathbf{A}_t^T\mathbf{A}_t + \gamma\mathbf{I}_d = \mathbf{H}_t$). Note that, unlike Section 4.1, the coherence of $\mathbf{A}_t$, denote $\mu_t$, changes with iterations.

**Theorem 2.** *Let $\mu_t \in [1, n/d]$ be the coherence of $\mathbf{A}_t$ and $m$ be the number of partitions. Assume the local sample size satisfies $s_t \geq \frac{3\mu_t d}{\eta^2}\log\frac{md}{\delta}$ for some $\eta, \delta \in (0, 1)$. Under Assumption 1, it holds with probability $1 - \delta$ that*

$$\left\|\boldsymbol{\Delta}_{t+1}\right\|_2 \leq \max\left\{\alpha\sqrt{\frac{\sigma_{\max}(\mathbf{H}_t)}{\sigma_{\min}(\mathbf{H}_t)}}\|\boldsymbol{\Delta}_t\|_2, \frac{2L}{\sigma_{\min}(\mathbf{H}_t)}\|\boldsymbol{\Delta}_t\|_2^2\right\},$$

*where $\alpha = \frac{\eta}{\sqrt{m}} + \eta^2$.*

**Remark 3.** *The standard Newton's method is well known to have local quadratic convergence; the quadratic term in Theorem 2 is the same as Newton's method. The quadratic term is caused by the non-quadritic objective function. The linear term arises from the Hessian approximation. For large sample size, $s$, equivalently, small $\eta$, the linear term is small.*

Note that in Theorem 2 the convergence depends on the condition numbers of the Hessian at every point. Due to the Lipschitz assumption on the Hessian, it is easy to see that the condition number of the Hessian in a neighborhood of $\mathbf{w}^\star$ is close to $\kappa(\mathbf{H}^\star)$. This simple observation implies Corollary 3, in which the dependence of the local convergence of GIANT on iterations via $\mathbf{H}_t$ is removed.

**Assumption 2.** *Assume $\mathbf{w}_t$ is close to $\mathbf{w}^\star$ in that $\|\boldsymbol{\Delta}_t\|_2 \leq \frac{3}{L} \cdot \sigma_{\min}(\mathbf{H}^\star)$, where $L$ is defined in Assumption 1.*

**Corollary 3.** *Under the same setting as Theorem 2 and Assumption 2, it holds with probability $1 - \delta$ that*

$$\left\|\boldsymbol{\Delta}_{t+1}\right\|_2 \leq \max\left\{2\alpha\sqrt{\kappa}\|\boldsymbol{\Delta}_t\|_2, \frac{3L}{\sigma_{\min}(\mathbf{H}^\star)}\|\boldsymbol{\Delta}_t\|_2^2\right\},$$

*where $\kappa$ is the condition number of the Hessian matrix at $\mathbf{w}^\star$.*

## 4.3 Inexact Solutions to Local Sub-Problems

In the $t$-th iteration, the $i$-th worker locally computes $\tilde{\mathbf{p}}_{t,i}$ by solving $\widetilde{\mathbf{H}}_{t,i}\mathbf{p} = \mathbf{g}_t$, where $\widetilde{\mathbf{H}}_{t,i}$ is the $i$-th local Hessian matrix defined in (5). In high-dimensional problems, say $d \geq 10^4$, the exact formation of $\widetilde{\mathbf{H}}_{t,i} \in \mathbb{R}^{d \times d}$ and its inversion are impractical. Instead, we could employ iterative linear system solvers, such as CG, to inexactly solve the arising linear system in (6). Let $\tilde{\mathbf{p}}'_{t,i}$ be an inexact solution which is close to $\tilde{\mathbf{p}}_{t,i} \triangleq \widetilde{\mathbf{H}}_{t,i}^{-1} \mathbf{g}_t$, in the sense that

$$\left\| \widetilde{\mathbf{H}}_{t,i}^{1/2} \left( \tilde{\mathbf{p}}'_{t,i} - \tilde{\mathbf{p}}_{t,i} \right) \right\|_2 \leq \frac{\epsilon_0}{2} \left\| \widetilde{\mathbf{H}}_{t,i}^{1/2} \tilde{\mathbf{p}}_{t,i} \right\|_2, \tag{9}$$

for some $\epsilon_0 \in (0,1)$. GIANT then takes $\tilde{\mathbf{p}}'_t = \frac{1}{m} \sum_{i=1}^{m} \tilde{\mathbf{p}}'_{t,i}$ as the approximate Newton direction in lieu of $\tilde{\mathbf{p}}_t$. In this case, as long as $\epsilon_0$ is of the same order as $\frac{\eta}{\sqrt{m}} + \eta^2$, the convergence rate of such inexact variant of GIANT remains similar to the exact algorithm in which the local linear system is solved exactly. Theorem 4 makes convergence properties of inexact GIANT more explicit.

**Theorem 4.** *Suppose inexact local solution to* (6)*, denote $\tilde{\mathbf{p}}'_{t,i}$, satisfies* (9)*. Then Theorems 1 and 2 and Corollary 3 all continue to hold with* $\alpha = \left( \frac{\eta}{\sqrt{m}} + \eta^2 \right) + \epsilon_0$.

Proposition 5 gives conditions to guarantee (9), which is, in turn, required for Theorem 4.

**Proposition 5.** *To compute an inexact local Newton direction from the sub-problem* (6)*, suppose each worker performs*

$$q = \log \frac{8}{\epsilon_0^2} \Big/ \log \frac{\sqrt{\tilde{\kappa}_t} + 1}{\sqrt{\tilde{\kappa}_t} - 1} \approx \frac{\sqrt{\kappa_t} - 1}{2} \log \frac{8}{\epsilon_0^2}$$

*iterations of CG, initialized at zero, where $\tilde{\kappa}_t$ and $\kappa_t$ are, respectiely, the condition number of $\widetilde{\mathbf{H}}_{t,i}$ and $\mathbf{H}_t$. Then requirement* (9) *is satisfied.*

## 5 A Summary of the Empirical Study

Due to the page limit, the experiments are not included in this paper; please refer to the long version [41] for the experiments. The Apache Spark code is available at https://github.com/wangshusen/SparkGiant.git. Here we briefly describe our results.

We implement GIANT, Accelerated Gradient Descent (AGD) [23], Limited memory BFGS (L-BFGS) [12], and Distributed Approximate NEwton (DANE) [36] in Scala and Apache Spark [44]. We empirically study the $\ell_2$-regularized logistic regression problem (which satisfies our assumptions):

$$\min_{\mathbf{w}} \frac{1}{n} \sum_{j=1}^{n} \log \left( 1 + \exp(-y_j \mathbf{x}_j^T \mathbf{w}) \right) + \frac{\gamma}{2} \|\mathbf{w}\|_2^2, \tag{10}$$

We conduct large-scale experiments on the Cori Supercomputer maintained by NERSC, a Cray XC40 system with 1632 compute nodes, each of which has two 2.3GHz 16-core Haswell processors and 128GB of DRAM. We use up to 375 nodes (12,000 CPU cores).

To apply logistic regression, we use three binary classification datasets: MNIST8M (digit "4" versus "9", thus $n = 2M$ and $d = 784$), Covtype ($n = 581K$ and $d = 54$), and Epsilon ($n = 500K$ and $d = 2K$), which are available at the LIBSVM website. We randomly hold $80\%$ for training and the rest for test. To increase the size of the data, we generate $10^4$ random Fourier features [26] and use them in lieu of the original features in the logistic regression problem.

For the four methods, we use different settings of the parameters and report the best convergence curve; we do not count the cost of parameter tuning. (This actually favors AGD and DANE because they have more tuning parameters than GIANT and L-BFGS.) Using the same amount of wall-clock time, **GIANT consistently converges faster than AGD, DANE, and L-BFGS** in terms of both training objective value and test classification error (see the figures in [41]).

Our theory requires the local sample size $s = \frac{n}{m}$ to be larger than $d$. But in practice, GIANT converges even if $s$ is smaller than $d$. In this set of experiments, we set $m = 89$, and thus $s$ is about half of $d$. Nevetheless, GIANT converges in all of our experiments. Our empirical may imply that the theoretical sample complexity can be potentially improved.

We further use data augmentation (i.e., adding random noise to the feature vectors) to increase $n$ to 5 and 25 times larger. In this way, the feature matrices are all dense, and the largest feature matrix we use is about 1TB. As we increase both $n$ and the number of compute nodes, the advantage of GIANT further increases, which means GIANT is more scalable than the compared methods. It is because as we increase the number of samples and the number of nodes by the same factor, the local computation remains the same, but the communication and synchronization costs increase, which favors the communication-efficient methods; see the figures and explanations in [41].

## 6    Conclusions and Future Work

We have proposed GIANT, a practical Newton-type method, for empirical risk minimization in distributed computing environments. In comparison to similar methods, GIANT has three desirable advantages. First, GIANT is guaranteed to converge to high precision in a small number of iterations, provided that the number of training samples, $n$, is sufficiently large, relative to $dm$, where $d$ is the number of features and $m$ is the number of partitions. Second, GIANT is very communication efficient in that each iteration requires four or six rounds of communications, each with a complexity of merely $\tilde{\mathcal{O}}(d)$. Third, in contrast to all other alternates, GIANT is easy to use, as it involves tuning one parameter. Empirical studies also showed the superior performance of GIANT as compared several other methods.

GIANT has been developed only for unconstrained problems with smooth and strongly convex objective function. However, we believe that similar ideas can be naturally extended to *projected Newton* for constrained problems, *proximal Newton* for non-smooth regularization, and *trust-region method* for nonconvex problems. However, strong convergence bounds of the extensions appear nontrivial and will be left for future work.

## Acknowledgement

We thank Kimon Fountoulakis, Alex Gittens, Jey Kottalam, Zirui Liu, Hao Ren, Sathiya Selvaraj, Zebang Shen, and Haishan Ye for their helpful suggestions. The four authors would like to acknowledge ARO, DARPA, Cray, and NSF for providing partial support of this work. Farbod Roosta-Khorasani was partially supported by the Australian Research Council through a Discovery Early Career Researcher Award (DE180100923).

## Footnotes

[1]As for general convex problems, it is very hard to present the comparison in an easily understanding way. This is why we do not compare the convergence for the general convex optimization.

[2]The second-order methods typically have the local convergence issue. Global convergence of GIANT can be trivially established by following [32], however, the convergence rate is not very interesting, as it is worse than the first-order methods.

[3]If the samples themselves are i.i.d. drawn from some distribution, then a data-independent partition is equivalent to uniform sampling. Otherwise, the system can Shuffle the data.

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
