[Reviews · NeurIPS 2018]

Reviewer 1



This paper proposes a distributed and communication-efficient Newton-type optimization method for the empirical risk minimization problem in distributed computing environment. At every iteration, each worker locally finds an Approximate NewTon (ANT) direction and send to the main driver, which averages all the ANT directions received from workers to form a Globally Improved ANT (GIANT) direction. The authors theoretically show that GIANT enjoys an improved convergence rate as compared with first-order methods and existing distributed Newton-type methods. Besides, a highly advantageous practical feature of GIANT is that it only involves one tuning parameter. Large-scale experiments on a computer cluster empirically demonstrate the superior performance of GIANT. Overall, the presentation of this paper is good. The related works and different research pesperctives for distributed learning are well overviewed and organized . While I'm not an expert in the distributed learning field, I have no difficulty following the paper. Through reading this paper, I feel like that I have catched one rough map of this field. From my understanding, the communication efficiency mainly comes from using the harmonic mean to approximate the arithmetic mean. The superior theorectical convergence results are mainly due to the matrix incoherence assumption and analysis . I think the incoherent data assumption is resonable. My main question is that, as the incoherence assumption is common in matrix completion and approximation, is this rarely used in distributed Newton-type optimization? If such kind of work exists, review of them and discussion of the difference between this paper are suggested.

Reviewer 2



This paper proposes a communication efficient distributed approximation to the Newton method called GIANT). The key idea is to approximate the global Hessian matrix by the local Hessian matrix in the worker nodes and to send the approximate Newton direction instead of the local Hessian matrix itself from the worker nodes to the driver to reduce the communication cost. For quadratic objective functions, the number of iterations of GIANT, and hence the communication complexity, is logarithmically dependent on the condition number of the Hessian matrix. The given analysis not only shows that GIANT is overall more communication efficient than state-of-the-art distributed methods but even converges in fewer iterations. This is also confirmed experimentally in a wide variety of settings. Also, in contrast to existing methods, GIANT has only one tuning parameter. Clearly, the topic of the paper is of high interest to the NIPS community. To the best of my knowledge, this is the first work to propose or at least to carefully analyze the properties of sending only approximate Newton directions. Also, the paper is well written and organized (One minor point: theorem numberings in Appendix do not have one-one correspondence to the theorems in the main paper). Weak points are that the experimental evaluation only uses two datasets. Increasing this seems particularly relevant in order to demonstrate the practicality of the technical assumption of incoherence. Also, the implementation seems to be not publicly available, which hinders reproducibility.

Reviewer 3



The paper introduces GIANT, a distributed variant of Newton algorithm. The considered problem is important and the paper gives a nice contribution to the field of distributed optimisation. The paper is very clear and nice to read, and propose nice theoretical contributions and experiments, with a detailed bibliography and positioning with respect to priori work. Here is my main criticism : * Authors acknowledge that their approach is close to previous works, namely DANE, for which GIANT seem to coincide to DANE in the least-squares loss case. However, the rate obtained in the paper is much better, certainly thanks to the introduction of the incoherence assumption, which is well known in the field of compressed sensing and randomized linear algebra. I think that the paper would benefit from a more thorough comparison of approaches, and to what extent this row-coherence assumption is reasonable in practical assumptions. It would help also to understand where the gap between the rate of GIANT and DANE comes from, if it is either related to the different set of assumptions, or if it comes from an alternative proof technique. * The numerical experiments are a bit disappointing. It proposes a comparison to good baselines, but on medium scale datasets (epsilon, covertype) and then on larger datasets, constructed as extensions of cover type. Many large or huge datasets for binary classification are available : it would be really interesting and more transparent to see the performances of these distributed optimisation algorithms for more realistic problems, for which feature matrices are really different. Indeed, considering different simulation setups for features hardly accounts for the variability of features in realistic problems. Indeed, the process of augmentation of the cover type dataset is probably helping with the incoherence, since it consists in row duplications with small Gaussian noise and features are augmented with random Fourier features. Furthermore, no experiments / guarantees for sparse features matrices are provided. In big data setting, with very large n / large d, features are typically sparse (word counts for instance) In summary, I like the paper and think that it deserves to be published, but I won't be shocked if it is not. I sincerely believe that it can be strongly improved by an extension of the experiments to several real-world large scale datasets, that are now commonly available in kaggle or UCI platforms.